# The LJ System—Development and Validation of a Reliable and Simple Device for Bronchoscopic Measurement of Distances Within the Tracheobronchial Tree

**DOI:** 10.3390/diagnostics15080954

**Published:** 2025-04-09

**Authors:** Zuzana Sestakova, Sarka Nemcova, Petr Denk, Veronika Pechkova, Pavel Michalek, Jiri Votruba

**Affiliations:** 1First Department of Tuberculosis and Respiratory Diseases, General University Hospital in Prague, 128 08 Prague, Czech Republic; veronika.pechkova@vfn.cz (V.P.); jiri.votruba@vfn.cz (J.V.); 2Faculty of Mechanical Engineering, Czech Technical University in Prague, 166 07 Prague, Czech Republic; sarka.nemcova@fs.cvut.cz (S.N.); petr.denk@fs.cvut.cz (P.D.); 3Department of Anesthesiology and Intensive Medicine, General University Hospital, First Faculty of Medicine, Charles University in Prague, 128 08 Prague, Czech Republic; pavel.michalek@vfn.cz

**Keywords:** airway distance measurement, interventional bronchology, measurement device, stent placement

## Abstract

**Background:** The accurate measurement of the distances within the airways during bronchoscopy is necessary for diagnostic purposes; however, a reliable and simple device does not exist. **Methods:** The LJ system, consisting of a probe, a box with a display, an encoder, and a microcontroller, has been developed, and its prototype has been tested in vitro and validated in clinical practice in suitable procedures of interventional bronchoscopy. **Results:** In vitro, the device measurements showed a good correlation with the control performed with a digital caliper. Subsequently, ten patients were included in a pilot study evaluating this novel prototype of a measurement device. The device was used on four patients with tracheal stenosis indicated for Y-stent placement, four patients indicated for open surgery, and two cases of tracheoesophageal fistula. The measurements have been validated using computed tomography imaging or by direct inspection and measurement during open surgical procedures. **Conclusions:** The first experience and pilot study evaluating this novel instrument for distance measurements during interventional bronchology procedures showed that the LJ device can provide precise readings of the distance from the vocal cords, the lengths of tracheal stenoses, or the size of tumorous and other lesions. Its use might be widened to other endoscopic indications.

## 1. Introduction

The precise measurement of the distances within the tracheobronchial tree can play a principal role in diagnostic and invasive bronchoscopy procedures. Such measurements have been technically difficult to execute so far, as the principle of flexible endoscope usage makes precise endobronchial measurement very challenging [1]. There have been attempts to measure tracheal and bronchial distances by optical means and, recently, by the use of electromagnetic navigation or robotic endoscopy. Knowledge of the lengths of the trachea and bronchi, as well as distances from bronchial orifices to lesions, can be one of the crucial pieces of information in bronchology [2]. Precise distance measurement inside the tracheobronchial tree may potentially be used in different sub-specialities, as follows: in interventional bronchology for the exact planning of endobronchial straight- and Y-stent insertions, for the ideal choice of endobronchial valves, for finding the right distance from the bronchial orifice to the target lesion during diagnostic endoscopy [3], and for an accurate double-lumen tube cannula placement in anesthesiology [4], as well as providing important information for the surgeon in various clinical scenarios of thoracic surgery [5].

Therefore, we have decided to develop and validate a simple and reliable device that enables us to measure endobronchial and extrabronchial distances. The instrument, however, can be used in many other clinical conditions, including stent placements and measurements of stenotic areas in the airways.

## 2. Materials and Methods

The first phase involved the development and in vitro evaluation of the measuring instrument, while the second phase, after the Ethical Committee approval, consisted of a pilot evaluation of the device in selected cases of interventional bronchology clinical practice.

### 2.1. Development of Measuring Tool

We designed and manufactured a tool that measures the depth of the probe, i.e., how far the tip of the probe is from the edge of the distal end of the bronchoscope (Figure 1). The bronchoscope used for all in vitro and in vivo experiments was a flexible bronchoscope Fujifilm EB-580T (Fujifilm, Tokio, Japan). The desired parameters and features were as follows:A lightweight box attachable to the bronchoscope handpiece;The measurement of the probe depth in the range of 200 mm, with an accuracy of 0.5 mm;A manually movable probe;Compatible with probe diameters from 1.1 mm to 2.2 mm.
Figure 1The mechanical design of the distance-measuring instrument LJ. The display, main switch, cantilever spring, batteries, and reset button are not included in the image.
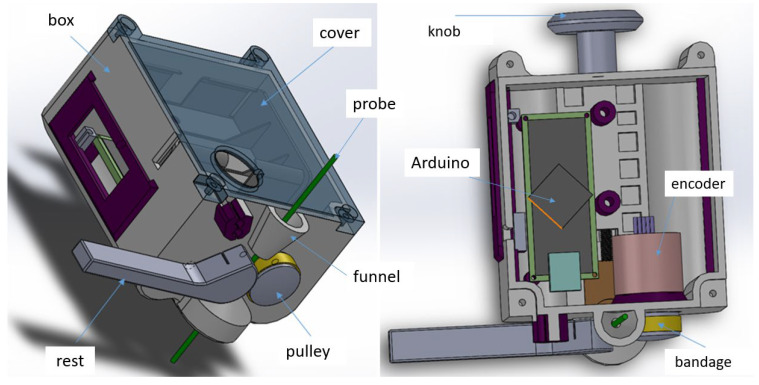


The measuring principle is based on the mechanical transmission of the linear movement of the probe to a rotary motion of a pulley attached to an encoder. The probe passes between the pulley and the mechanical rest with a guide groove, which is pressed against the pulley with a cantilever spring (made from a joint gauge). The encoder’s output is read using an Arduino Nano microcontroller. The signal is processed, converted into a distance in millimeters, and displayed on a graphical OLED display. The user can set the depth to zero by pressing the reset button when the probe is just at the edge of the distal end, which is visualized with the camera of the bronchoscope.

First, we had to find a suitable combination of the pulley diameter and the encoder’s number of pulses. From the number of pulses and the required reading accuracy, the pulley diameter can be calculated (see Table 1).

We chose the Megatron MOM18 360 513 BZ encoder with 400 pulses per revolution to ensure that the accuracy was met. The pulley was provided with a silicone bandage, so as to prevent the probe from sliding on the pulley and causing an incorrect reading of the probe depth. We performed an experiment to find the sufficient downforce that the mechanical rest must provide for reliable transmission of the probe shift to the pulley rotation. We designed the cantilever spring according to the results of this experiment. To aim the probe between the pulley and the rest, there is a funnel guide above the pulley. The design of the rest allows the user to release it from the pulley so that there is a gap through which the delicate end of a piezo probe can be safely pushed.

Most parts of the measuring instrument were made by rapid prototyping and printed using a stereolithography (resin) 3D printer [6]. The design was made considering the need for proper cleaning. For example, the number of through holes was minimized. The dimensions and weight were also kept low, so that the instrument was not in the way when the user held the bronchoscope’s handpiece. The method of connecting the instrument onto the bronchoscope was designed to be easy and simple to use (Figure 2). There is a bore with a spring-loaded, U-shaped part that connects to the metal pin on the handpiece, which is the entry point for the probe. To attach the instrument to the bronchoscope, the user just pulls the knob at the back of the box, puts the box onto the pin, and releases the knob. The instrument is held securely on the handpiece. As it is battery-powered, there are no cables. The whole box can be cleaned by wiping it with a wet cloth.

### 2.2. In Vitro Evaluation

The first phase of the device accuracy evaluation involved the measurement of distances within the tracheobronchial tree on an airway simulator, using a flexible bronchoscope Fujifilm EB-580T. An AirSim Bronchi (TruCorp Ltd., Craigavon, Co., Armagh, UK) airway manikin was used for all of these experiments. Ten different measurements were performed with the LJ system prototype on the airway manikin, and they included the following:Measurement of the distance between the vocal cords and a foreign body inserted into the trachea;Measurement of the length of the foreign body inserted into the trachea;Measurement of the distance between the vocal cords and the incision made to the tracheal wall;Measurement of the length of tracheal stenosis simulated with a clip placed externally on the manikin’s trachea;Measurement of the distance between the main carina and the orifice of the right main bronchus.

All measurements performed with the LJ system were recorded and subsequently compared with direct measurements on the trachea using a digital caliper (Kinex Iconic, Kinex, Czech Republic). The null hypothesis was that there was no statistical difference between the mean values of both measurements, while the alternative hypothesis was that the values significantly differed (*p* ≤ 0.05). The corresponding values and their differences were subsequently compared using the Student paired T-test.

### 2.3. Pilot Clinical Evaluation

For the human part of the pilot study, ethical approval from a local Ethical Committee (General University Hospital in Prague, date: 21 January 2021, no. 1790/20 S-IV) was obtained prior to the enrolment of the first patient. All patients received a written Patient Information Sheet (PIS) prior to enrolment and signed a consent form.

All interventional bronchologic examinations were performed at the interventional bronchology suite under general anesthesia. Before starting the clinical case, the LJ device was connected to the flexible bronchoscope (Fujifilm EB-580T), as shown in Figure 2. After approaching the beginning of pathology, or the predetermined starting point of measurement, the device was activated. At the end of the measurement, the distance in mm was retrieved from the display of the LJ device and recorded on the Case Report Form (CRF).

## 3. Results

Ten different measurements were performed on the manikin. The results are expressed in Table 2.

Freeware R (version 2023, R Computing, Vienna, Austria) was used for statistical analysis. The difference in the mean values was −0.36 mm, with a 95% confidence interval of −0.884 to 0.165 mm. The *p* value was 0.155 and, therefore, the null hypothesis was confirmed, and the mean values were considered statistically not different. The difference of 1 to 1.4 mm (items 2, 3, and 6) may be due to the probe being moved too rapidly, causing pulse losses. The shape of the rest head, which is in contact with the probe, will be redesigned to ensure better adhesion of the probe to the pulley.

In the clinical part of the pilot study, in total, ten clinical tests were performed with the measurements within the airway using the LJ device. Their detailed description is provided in Table 3. Five measurements (50%) were performed for subsequent interventional bronchologic procedures, mainly for the determination of the stent length. Another four measurements (40%) were taken before the planned open surgical procedures to confirm the extent of the procedure. The remaining measurement (10%) was performed solely for diagnostic purposes. Some of the most interesting scenarios are described in detail and illustrated below.

Case study 1

A 70-year-old, disabled pensioner, who was an ex-smoker with a history of 40-pack-years, after four cycles of chemoimmunotherapy for central lung cancer originating from the right lung (histologically determined as non-small-cell lung carcinoma with liver metastasis) was admitted to the inpatient pulmonology department as an acute admission because of shortness of breath progression, accompanied by respiratory insufficiency.

Clinically, exertional dyspnea dominated, which had worsened over the previous few days, as well as a cough, general weakness, fatigue, and a loss of appetite. In the physical examination, the patient had an oxygen saturation on pulse oximetry of 88%, was normotensive, without tachycardia, and had a BMI of 32 kg·m^−2^. His breathing was weakened basally on both sides, without secondary phenomena. The laboratory examination excluded inflammatory etiology. Complementary CT-angio scanning ruled out pulmonary embolism but showed the progression of a tumorous mass under the right hilum, with compression of both main bronchi and growth of the tumor into the left main bronchus (Figure 3).

Palliative rigid bronchoscopy was indicated, with the aim of re-opening the central airways. During bronchoscopy, tumorous granulations were visualized on the main carina, which completely occluded the left-sided bronchial tree. On the right side, the main bronchus and bronchus intermedius were funnel-shaped, the hilar carina was significantly widened, and the orifice of the right upper lobe bronchus was free. The area around the middle bronchus orifice was covered with fragile contact bleeding granulations, and the lumen of the lower lobe bronchus on the right was completely obliterated.

After mechanical disobliteration of tumor masses using a cryoprobe, it was possible to insert a Y stent. We adjusted the Dumon stent (16 × 30 × 30) according to the distance measurements using the LJ system (Figure 4). Subsequently, a “stent into stent” (Microtech t 12 × 20) was inserted into the right lower bronchus for torsional changes, twisting the distal part of the right arm of the original Dumon stent.

After the procedure, the patient developed slight hemoptysis and difficult expectoration, with the need for intensive mucolytic therapy and bronchoscopic toileting. In the further course, the clinical condition improved, including oxygen saturation, and the patient was able to continue with chemoimmunotherapy.

Case study 2

A 37-year-old patient, who was quadriplegic as a result of tick-borne meningoencephalitis, with a long-term tracheostomy, tracheal stenosis, and dependence on full-time nursing care, was indicated for consideration of the removal of the tracheostomy cannula. We used the LJ device to accurately measure the distance of the stenosis from the vocal cords and the length of the stenosis inside the trachea. We measured the distance of the stenosis from the cords to 2.9 cm, and we also quantified the length of the stenosis in the trachea to 1.5 cm (Figure 5). Based on these findings and a multidisciplinary board decision, we indicated this patient for end-to-end tracheal resection with reconstruction of the stenotic area.

## 4. Conclusions

A novel device for the bedside measurement of distances within the trachea and the main bronchi was developed and subsequently validated on an in vitro model and on the series of intraluminal airway pathologies. The measurements obtained on an airway manikin, which were subsequently checked with a digital caliper, showed high accuracy and justified the use of the device in clinical practice. The series of real pathologies within the airway was measured with the LJ device, and it helped to establish a subsequent strategy for either endoscopic or surgical treatment. The device was very easy to use, light, and resistant to any accidental damage. Any probe can be used as a measuring tool, including a cryoprobe, flexible forceps, or a rEBUS probe, which we perceive as a substantial advantage of the system. The measurements obtained using the LJ device were also repeatable and provided identical results when evaluating the same distances within the airway below the level of the vocal cords.

The precise measurement of the distances within the tracheobronchial tree is crucial for subsequent surgical or interventional bronchologic procedures. However, only a few reports in the literature exist to address this problem. The simplest, but not always precise, method is to use a flexible bronchoscope and place a mark, usually with a piece of plastic tape, on the bronchoscope when reaching the distal part of a measured distance. The bronchoscope is then withdrawn to the beginning of the measured distance and the final value is quantified on the bronchoscope using a ruler or caliper. This technique has been used for the measurement of the distance between the incisors and the main carina in double-lumen tube placement for open thoracic surgery [7]. A similar technique was also employed for the estimation of the ideal depth of tracheal tube insertion in adult patients undergoing general anesthesia [8]. Some types of computer software are able to estimate the distances from the images taken with a fiberscope [9].

Other methods described in the literature use processes based on CT scanning. They are usually very accurate, but not applicable as a bedside method, and are also relatively expensive. Morshed et al. compared CT scanning with multiplanar re-formatting, bronchoscopic evaluation, and direct intraoperative measurement [10]. They found that CT-based virtual endoscopy has higher sensitivity and identical specificity when compared with multiplanar CT re-formatting. Li et al. performed CT scanning combined with flexible bronchoscopy for the assessment of intraluminal airway distances in order to correctly insert a right-sided double-lumen tube [11]. Three-dimensional CT imaging was used to measure the length of the trachea in 215 patients indicated for an open chest procedure [12]. Fluoroscopy augmented with a cone-beam CT scan can be utilized for the measurement of distances of peripheral lung lesions in a hybrid operating room [13]. A high-frequency ultrasound probe is another method feasible for the evaluation of distances within the major airway. It showed a high correlation with bronchoscopic measurements and CT scan findings in one study [14]. Endobronchial ultrasound (EBUS) has been repeatedly used for the measurement of distances in peripheral pulmonary nodules and other lesions [15,16]. Various authors have performed measurements with a radial probe [17] or radial EBUS combined with transbronchial lung biopsy [18]. Several methods describing the measurement of the airway diameter and cross-sectional areas of the tracheobronchial tree have also been published. A laser probe introduced through the working channel of a flexible bronchoscope was used in one study to measure the diameter of the trachea and main bronchi [19]. The results were reproduced with special software created by the National Institute of Health. Williamson et al. used anatomical optical coherence tomography (aOCT) to measure airway diameters and areas [20]. The optical coherence tomography unit is composed of a special thin probe inserted into the working channel of any flexible bronchoscope. The probe can emit a broadband light beam at a wavelength of 1310 nm perpendicular to the tracheobronchial tree, and the signal is transmitted to the monitor; in addition, software enables the quantification of diameters and areas within the tube. The authors validated their device on a plastic tube, an animal (pig) model, and a pilot series of patients scheduled for bronchoscopy. The results showed a very good correlation with the control performed using CT scanning.

This study has several limitations. First, because of the pilot design, the number of subjects is very small. This study describes a preliminary experience, and there is no comparison group. Second, the results of measurements in clinical cases have not been consistently compared with CT findings or another method. We plan to assess the correlations of the measurements obtained with the LJ device with CT imaging in a future prospective study. Also, the possible difference between rigid and flexible endobronchial measurements has not been studied yet. Validation in the next cases will be performed using rigid bronchoscopy. All of the measurements were taken in patients under general anesthesia. In reality, similar measurements are most often taken under light sedation. We are currently working on the elucidation of the influence of the type of sedation/anesthesia on measurement results.

The device used in this project was designed as a proof-of-concept prototype. Its cost was approximately EUR 400, including an off-the-shelf encoder, various other electronics, in-house manufactured and standard metal parts, and in-house 3D resin printing. To ensure a sterile device, the final device will be disposable, therefore, some design modifications will be made.

In conclusion, this novel LJ device may become a useful tool for interventional bronchologists and other specialists involved in airway procedures. Its principles can be also utilized in other disciplines employing the measurement of distances during endoscopic examinations, such as esophagoscopy, gastroscopy, colonoscopy, or cystoscopy.

## Figures and Tables

**Figure 2 diagnostics-15-00954-f002:**
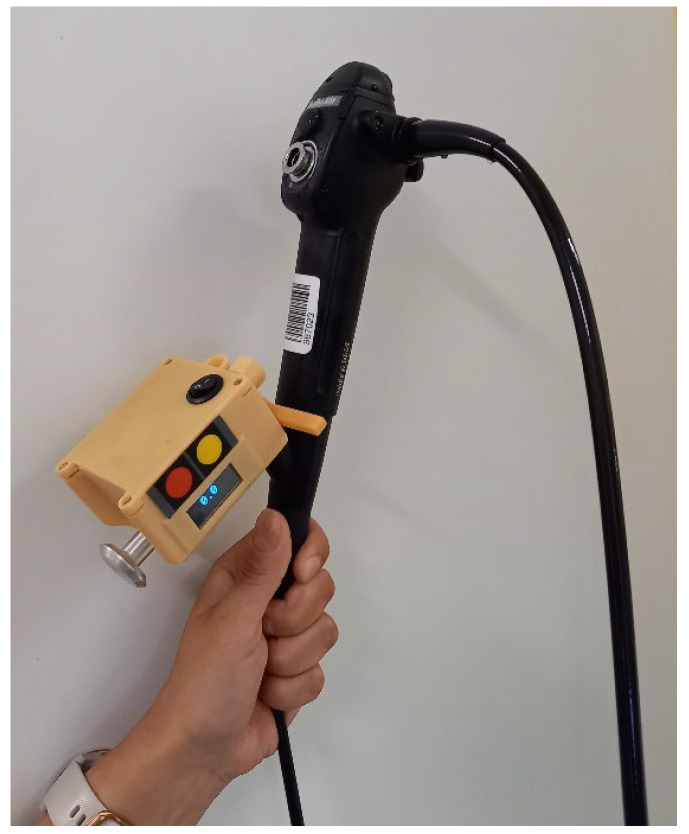
The LJ system attached to the flexible bronchoscope.

**Figure 3 diagnostics-15-00954-f003:**
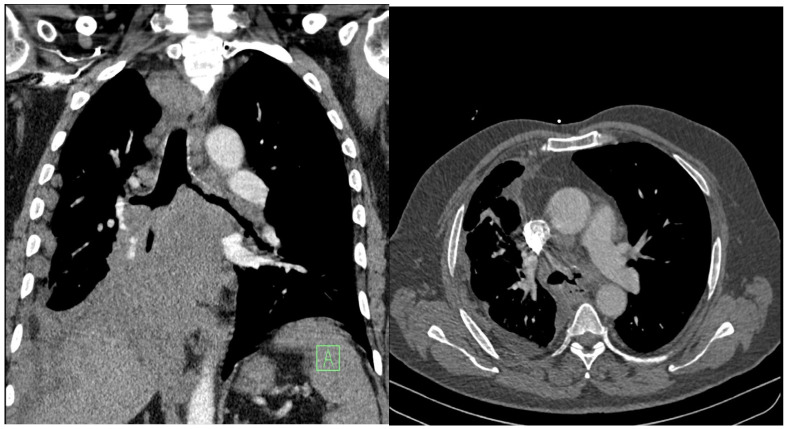
Progression of the central tumor on the right side of the lungs. Vascular structures and the main bronchi are involved and compressed, and the masses also extend into the left main bronchus.

**Figure 4 diagnostics-15-00954-f004:**
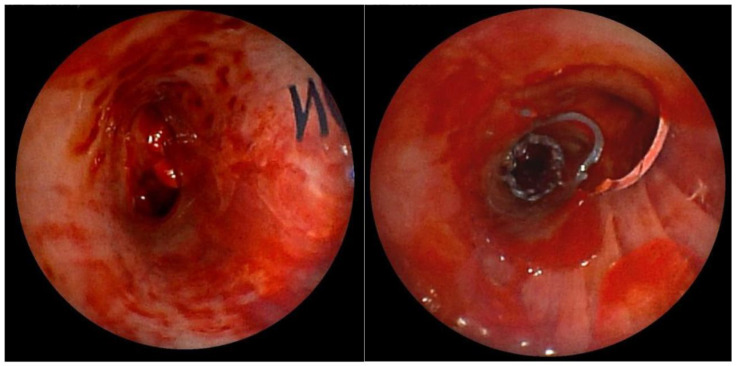
Left leg of the Dumon stent (**left**) and right arm of the stent with a cut-out hole for the upper lobar bronchus. In the distal part of the stent, a Microtech “stent into the stent” was inserted for torsional changes (**right**).

**Figure 5 diagnostics-15-00954-f005:**
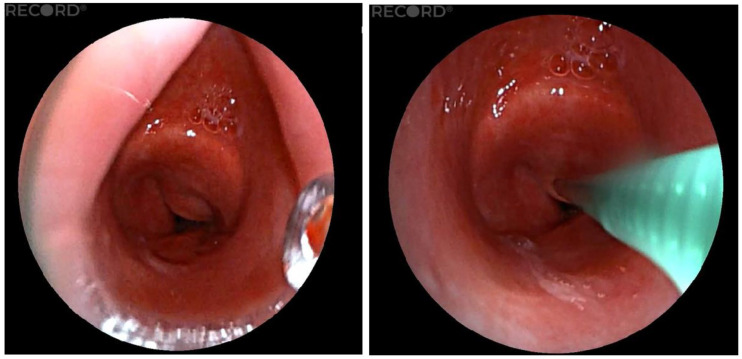
Measurement of the length of the stenotic portion of the trachea.

**Table 1 diagnostics-15-00954-t001:** Calculated pulley diameters according to the encoder’s number of pulses per revolution.

Reading Accuracy [mm]	Encoder’s Pulses/Revolution	Diameter of Pulley [mm]
0.5	10	1.6
0.5	50	8
0.5	100	16

**Table 2 diagnostics-15-00954-t002:** The measurement of distances within the tracheobronchial tree in vitro on the manikin using the LJ device and a comparison with distances measured using a digital caliper.

Measurement	LJ Device (mm)	Digital Caliper (mm)
1	32.0	32.7
2	26.5	25.1
3	12.0	11.0
4	37.5	37.2
5	25.5	25.2
6	17.0	15.9
7	22.0	22.2
8	28.0	28.5
9	30.0	29.0
10	22.5	22.6

**Table 3 diagnostics-15-00954-t003:** Description of clinical cases with the measurement of distances within the airway using the LJ device.

Patient	Description
1	Measurement of distances for ideal Y stent size
2	Measurement of the length of tracheal stenosis for subsequent open resection
3	Measurement of the tracheoesophageal fistula extent for choosing a temporary stent
4	Measurement of the tumor spread from the hilar carina for surgical indication (sleeve lobectomy)
5	Measurement of the stent length in malignant tracheal stenosis
6	Stenosis measurement to facilitate the insertion of a hilar Y stent (right side)
7	Measurement of endobronchial tumor progression in bronchus intermedius to facilitate parenchyma-sparing surgery
8	Measurement of the stent length in malignant stenosis of the right main bronchus
9	Measurement of tumor infiltration length before planned surgical carinectomy in adenoid cystic carcinoma (both main bronchi measured)
10	Measurement of tracheoesophageal fistula orifice width and its distance to the main carina and vocal cords

## Data Availability

All data are available within the manuscript.

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
