# Peer review of "The LJ System—Development and Validation of a Reliable and Simple Device for Bronchoscopic Measurement of Distances Within the Tracheobronchial Tree"

_diagnostics, 2025, doi:10.3390/diagnostics15080954_

Round 1
Reviewer 1 Report
Comments and Suggestions for Authors
Review of the paper titled: “The LJ system – development and validation of a reliable and simple device for measurement of distances within the tracheo-3 bronchial tree”
This is an interesting paper describing a new tool for endoscopic precise measurement of the airway. The topic is interesting since a precise measurement represents the basis for planning the right treatment.
Tracheal resection needs to be planned after ac correct measuring of the airway disease, including distance from the vocal cords and from the main carina.
Similarly, a proper airway stenting needs to be planned with the right measures of the diseased airway.
Several authors support rigid bronchoscopy as the only tool able to get reliable airway measurement.
The authors propose a new digital device for airway measurement, to be used by flexible bronchoscopy, which seems to work.
I’ve few suggestions for them:
- Clearly this is a preliminary experience and there is no comparison group. These limits have been mentioned but must be outlined (and extended) separately in discussion.
- The title should include the message that the device is for flexible bronchoscopy.
- I didn't found any data about cost of the device. You should mention also this point.
- Validation in next cases should be considered by using rigid bronchoscopy. This point must be added in discussion.
English language must be reviewed (minor).
Author Response
1.Clearly this is a preliminary experience and there is no comparison group. These limits have been mentioned but must be outlined (and extended) separately in discussion.
We have extended the discussion in this context
- The title should include the message that the device is for flexible bronchoscopy.
The title has been changed
- I didn't found any data about cost of the device. You should mention also this point.
This has been added into the discussion
- Validation in next cases should be considered by using rigid bronchoscopy. This point must be added in discussion.
The discussion has been extended accordingly
Reviewer 2 Report
Comments and Suggestions for Authors
The article describes a new method of measuring the bronchial tree. The topic is very interesting and necessary.
My comments concern:
- what type of catheters were used for measurements in the bronchial tree?. It seems to me that ordinary bronchoscopic forceps were used (visible in the photo). It seems to me that any tool can be used as a measuring catheter, including a cryoprobe, e.g., during lung biopsy procedures. The authors must write this clearly. This is an advantage of the method and will certainly appeal to users
- I have doubts about the method of disinfection (disposable device?) of the part of the measuring system that is in contact with the probe. The authors write that disinfection is easy, but they do not provide details. The measuring device must be very thoroughly decontaminated before use, and ordinary cleaning is not sufficient
- the measurements were taken in patients under general anesthesia. In reality, similar measurements are most often taken under light sedation. During these procedures, the patient coughs. How would the readings behave in such a situation?
- during the first verification of measurement methods on phantoms in measurements 2, 3, and 6, the difference is one or more mm. How do the authors explain this?. Shouldn't the compliance be greater? - The article does not provide information on how the measuring probe will behave with a flex bronchoscope
Author Response
1.what type of catheters were used for measurements in the bronchial tree?. It seems to me that ordinary bronchoscopic forceps were used (visible in the photo). It seems to me that any tool can be used as a measuring catheter, including a cryoprobe, e.g., during lung biopsy procedures. The authors must write this clearly. This is an advantage of the method and will certainly appeal to users
We changed discussion part on this subject to be more clear.
- I have doubts about the method of disinfection (disposable device?) of the part of the measuring system that is in contact with the probe. The authors write that disinfection is easy, but they do not provide details. The measuring device must be very thoroughly decontaminated before use, and ordinary cleaning is not sufficient
Text on that subject has been added into discussion part.
3.the measurements were taken in patients under general anesthesia. In reality, similar measurements are most often taken under light sedation. During these procedures, the patient coughs. How would the readings behave in such a situation?
This has been added into discussion
4.during the first verification of measurement methods on phantoms in measurements 2, 3, and 6, the difference is one or more mm. How do the authors explain this?. Shouldn't the compliance be greater? - The article does not provide information on how the measuring probe will behave with a flex bronchoscope
The discussion has been extended accordingly
Round 2
Reviewer 1 Report
Comments and Suggestions for Authors
the paper was improved by revision and it is ready for publication